# Neurotrophins: Expression of Brain–Lung Axis Development

**DOI:** 10.3390/ijms24087089

**Published:** 2023-04-11

**Authors:** Sara Manti, Federica Xerra, Giulia Spoto, Ambra Butera, Eloisa Gitto, Gabriella Di Rosa, Antonio Gennaro Nicotera

**Affiliations:** 1Pediatric Unit, Department of Human and Pediatric Pathology “Gaetano Barresi”, AOUP G. Martino, University of Messina, Via Consolare Valeria, 1, 98124 Messina, Italy; 2Unit of Child Neurology and Psychiatry, Department of Human Pathology of the Adult and Developmental Age, “Gaetano Barresi” University of Messina, 98124 Messina, Italy; giulia.spoto27@gmail.com (G.S.); butera.ambra@gmail.com (A.B.); gdirosa@unime.it (G.D.R.); antonionicotera@ymail.com (A.G.N.); 3Intensive Pediatric Unit, Department of Human Pathology of the Adult and Developmental Age, “Gaetano Barresi” University of Messina, 98124 Messina, Italy; egitto@unime.it

**Keywords:** neurotrophins, neurotrophic factors, development, brain, lung

## Abstract

Neurotrophins (NTs) are a group of soluble growth factors with analogous structures and functions, identified initially as critical mediators of neuronal survival during development. Recently, the relevance of NTs has been confirmed by emerging clinical data showing that impaired NTs levels and functions are involved in the onset of neurological and pulmonary diseases. The alteration in NTs expression at the central and peripheral nervous system has been linked to neurodevelopmental disorders with an early onset and severe clinical manifestations, often named "synaptopathies" because of structural and functional synaptic plasticity abnormalities. NTs appear to be also involved in the physiology and pathophysiology of several airway diseases, neonatal lung diseases, allergic and inflammatory diseases, lung fibrosis, and even lung cancer. Moreover, they have also been detected in other peripheral tissues, including immune cells, epithelium, smooth muscle, fibroblasts, and vascular endothelium. This review aims to provide a comprehensive description of the NTs as important physiological and pathophysiological players in brain and lung development.

## 1. Introduction

Neurotrophins (NTs) are a group of soluble growth factors with analogous structures and functions, identified initially as critical mediators of neuronal survival during development. They include nerve growth factor (NGF), the first and best characterised NT, brain-derived neurotrophic factor (BDNF), neurotrophin-3 (NT-3), and neurotrophin-4 (NT-4). These proteins can regulate many aspects of neural functions in the nervous system and airway tree, modulating synaptic function, plasticity, neuronal survival, lung innervation, and inflammatory response, respectively [1,2,3,4].

NTs are produced by neuronal and non-neuronal cells: among the latter ones, in the central nervous system (CNS), in vivo studies confirmed the expression of BDNF by endothelial cells, astrocytes, and oligodendrocytes. Notably, the production of BDNF has been rated 50-fold higher in cerebral endothelial cells than in cortical neurons. However, the neurons present distinct regulated and constitutive secretory pathways, whereas the non-neuronal cells have only the constitutive pathway [5,6].

NTs are synthesised as preproprotein precursors and cleaved intracellularly by metalloproteases into pro-NTs. They are further processed to generate mature NTs, featured by extensive homology and structural similarities. Alternatively, NTs may be secreted in the pro-form and undergo extracellular cleavage into the smaller active peptides [4,7].

Recently, the relevance of NTs has been confirmed by emerging clinical data showing that impaired NTs levels and functions are involved in the onset of neurological and pulmonary diseases. Notably, NTs are involved in the embryo’s development from the early stages, and the disruption of the signalling pathways induced by these factors may result in several pathologies of the nervous and pulmonary systems [8,9].

This review aims to provide a comprehensive description of the NTs as important physiological and pathophysiological players in brain and lung development.

## 2. Neurotrophins: Role and Characteristics

NTs may exert their action through two different mechanisms: a genomic one, which is considered a slow pathway, and a non-genomic mechanism, which has a more rapid effect. By acting in a genomic manner, the NTs may regulate cell growth and survival and signalling events such as ionic fluxes (Figure 1). The term ‘non-genomic’ refers to the effects noted within short time frames and excludes subsequent genomic effects that may occur in response even to a brief stimulation with NTs [7].

Two different types of cell surface receptors are involved in the NTs-mediated genomic effects: p75 neurotrophin receptor (p75^NTR^) and Tropomyosin-related kinase (Trk) receptors [10,11]. Both p75^NTR^ and Trk receptors activate multiple and distinct signalling pathways [3,12]. The effects induced by these receptors are complex since they bind the NTs independently or interact with each other, producing different results [5].

p75^NTR^ is a 75 kDa receptor that binds all NTs with low affinity. It belongs to the tumour necrosis factor (TNF) receptor superfamily and, similar to other family members, contains a death domain that docks with intracellular adapter proteins. The interaction with p75^NTR^ activates a downstream signalling cascade involving transcriptional factors such as NFκB and leading to cellular apoptosis [1,2,13,14]. This process may require the expression of specific adaptor proteins as well as impaired Trk signalling and is the result of the interaction between p75^NTR^ and NTs in their immature form [15,16]. However, its primary function is to act as a coreceptor with Trks to create high-affinity NT receptors [15].

Trk receptors are a group of 140 kDa tyrosine kinases with greater specific affinity for the different NTs, preferentially in their mature forms. They are regulated by p75^NTR^ and require its presence to bind the NTs with high affinity: TrkA was first discovered as the primary signalling receptor for NGF, but in an alternatively spliced form, it can also bind to NT-3 with different effects; TrkB primarily interacts with BDNF and NT-4 and shows a lesser affinity for NT-3; and TrkC preferentially binds to NT-3 [5,15,17]. The NT/Trks binding results in the phosphorylation and pairing of intracellular tyrosine residues. This leads to rapid activation of downstream signalling cascades, such as extracellular-regulated kinase (Ras/MAPK/ERK), phosphoinositide-3 kinase/Akt (protein kinase B), and phospholipase Cγ pathways. These cascades activate cell-specific transcription factors responsible for differentiation and cell survival, apoptosis, and cell growth [17].

Regarding the “non-genomic” effect, NTs have relatively immediate effects and exert their action by binding high-affinity Trk receptors. Furthermore, NTs can directly modulate plasma membrane receptors (e.g., N-methyl-D-aspartate receptors), cationic channels (e.g., Ca^2+^ influx and voltage-gated Na^+^ channels), and other key mechanisms to neuronal function [1,2].

## 3. Neurotrophins and Brain

Since their discovery in 1951, several studies have demonstrated the role of NTs in the developing brain and the mature nervous system. They mediate cell migration and proliferation, synapse formation and function, and regulation of dendritic outgrowth and axonal orientation [18,19,20]. NTs are also involved in motor and cognitive functioning through direct interaction with different neurotransmitter systems: particularly, BDNF participates in the long-term potentiation by up-regulating the N-methyl-D-aspartate (NMDA) receptors, inducing the expression of Gamma-Aminobutyric Acid (GABA) receptor subunit genes, and facilitating the synthesis of choline acetyltransferase, allowing the production and release of acetylcholine (ACh) [8,20,21].

### 3.1. Neurotrophins and Brain Development

As previously discussed, p75^NTR^ may bind the NTs with lesser affinity or act as a coreceptor with Trks to create high-affinity NT receptors, leading to cell survival or death depending on the ligand [15]. When p75^NTR^ interacts with the immature forms of NTs, it induces proapoptotic signals. It has been demonstrated that the interaction pro-BDNF/p75^NTR^ exerts an opposite action to the antiapoptotic effect of BDNF/TrkB [22]. Finally, p75^NTR^ is also implicated in recycling the NTs and orienting the axon through axon-repulsive agents (e.g., myelin-associated glycoprotein) [19]. During the first trimester of pregnancy (8th–9th gestational weeks), a distinct group of heterogeneous post-migratory cells organizes in the so-called “cortical plate”, containing the neurons that will give rise to the cortical layers 2 to 6. The compartment of cells below the cortical plate constitutes the subplate and expresses markers of synaptic connectivity and axonal outgrowth. At the beginning of the third month of gestation (9th–10th weeks), a group of early-born glutamatergic neurons in the subplate zone expresses the p75^NTR^ and starts to extend subcortical axons [23].

Furthermore, Trk receptors have also been detected in several animal models at early stages of neuronal development: TrkC appears during neurulation in the neural plate of a chicken embryo, participating in neuronal migration. At the same time, TrkB has been identified in the neural tube and especially in the motor neuron progenitors [24]. The presence of NTs receptors since the first trimester of gestation indirectly supports the neurotrophic hypothesis. According to this assumption, in the early stages of development of the nervous system, there is an excess of neurons, some superfluous and inappropriately connected. As a consequence, neuronal survival or death is regulated by NTs produced in their target tissues [25].

Studies on NGF proved that the interaction between TrkA and NGF induces the activation of the Raf pathway, leading to axonal elongation [19,26]. Moreover, NGF/TrkA binding is implicated in neurite outgrowth, development and survival of cholinergic neurons in the CNS, and the survival and maintenance of peripheral neurons [27]. Therefore, during brain development, NGF is considered a critical growth factor for neuronal survival and especially the nociceptive phenotype of neurons [28].

On the other hand, BDNF is widely expressed in the CNS and represents the main regulator of its development and maturation since it is involved in neuronal growth, differentiation, and synaptic and structural plasticity [20]. In fact, BDNF-positive neurons in the frontal lobe increase in size during the fourth month of gestation [29]. Furthermore, it is implicated in establishing neurogenesis, maintaining brain homeostasis, and neuroprotection from the neurotoxic effects of inflammation [27]. It is also responsible, together with NT-3, for the regulation of angiogenesis and vessel maintenance during embryogenesis [19,20].

Conversely, NT-4 is the least studied factor among the NTs. Despite the overlapping distribution and receptor interaction with BDNF in the fetal nervous system, it has been suggested that NT-4 and BDNF may induce growth, differentiation, and cell death through different mechanisms. In addition, it has been proposed a role in the survival of motor and sensory neurons (especially the ones present in the taste buds of the tongue and palate), in the outgrowth of neurite ganglia in the retina, and synapse formation in the hippocampus [30].

There are sparse data in the literature regarding the role of NTs on the human nervous system development during pregnancy. NTs are known to be produced by the placenta and the fetal brain, and some authors hypothesised that especially BDNF and NT-3 participate in the preimplantation stage and early embryonic development [27]. Most of the research focused on BDNF for its involvement in the survival of CNS neurons during pregnancy. Many studies analysed its levels in the maternal blood and amniotic fluid during gestation to better understand its role in neurodevelopment. Indeed, studies on animal models demonstrated that BDNF levels in the maternal blood are consistent with the fetal brain ones [31].

Several hypotheses have been formulated on the synthesis and role of BDNF during gestation. Given that NTs can cross the fetoplacental barrier, Antonakopoulos et al. suggested that maternal BDNF reaches the fetal brain through the placenta and sustains brain maturation [29]. Accordingly, a longitudinal study showed a significant decline in serum BDNF levels in women from the first trimester to the second and from the second to the third. After the partum, serum BDNF levels significantly increase, reaching the previous ranges, thus proving the utilisation of maternal BDNF by the placenta and the fetus [31]. On the other hand, some authors reported increased levels of BDNF during the last trimester of pregnancy compared to the first trimester. This evidence led to the hypothesis of a combined production from the placenta and the fetus, being the BDNF mainly produced by the fetus during the second trimester [27,29]. BDNF levels have also been associated with the fetus’s growth impairment, with higher levels in intrauterine growth restriction and lower ones in maternal type 1 diabetes and nondiabetic macrosomia. Based on these data, it has been proposed that increased BDNF production may represent an adaptive response to an intrauterine adverse environment [27,29]. In addition, prematurity is also associated with reduced BDNF and NT-3 levels; therefore, lower neonatal BDNF levels have been suggested as an early marker of abnormal neurodevelopmental outcomes in preterm infants [20,27].

### 3.2. Neutrophins and Neurologic Diseases

For their role in neurogenesis, neuronal survival and differentiation, and synaptic function, the involvement of NTs in several neurological diseases is well established, both during the developmental age and adulthood (Table 1) [6,8,32,33,34,35,36,37].

Studies on hippocampal tissue of BDNF-knocked-out mice proved the implication of BDNF in synaptic plasticity and the development of a stable long-term potentiation, affecting learning and memory. Moreover, the BDNF/TrkB activation has been proposed as a mechanism of epileptogenesis promotion: BDNF induces the sprouting of the GABAergic mossy fibre of dentate granule cells, increasing seizure susceptibility. Accordingly, different studies proved that BDNF-knocked-out mice show a reduction in kindling, while transgenic mice that over-express BDNF present more severe seizures in response to kainic acid and spontaneous seizures [8].

In addition, the impairment of the synaptic plasticity of hippocampal GABAergic neurons caused by a reduction in BDNF/TrkB signalling has also been implicated in the development of Autism Spectrum Disorders (ASD) [36]. In the last few decades, several studies have analysed the involvement of NTs in Autism and other neurodevelopmental pathologies. Very recently, Gevezova et al. investigated the role of NGF in ASD and its possible use as a biomarker of mitochondrial function in these patients. The authors propose NGF as a regulator of biogenesis and homeostasis of the mitochondria, supporting its role in the aetiology and progression of ASD and other neurological diseases [37].

Recent studies on patients affected by schizophrenia have also detected a decrease in BDNF levels in CNS and peripheral serum samples. Accordingly, similar data were observed in several animal models of other neuropsychiatric disorders, such as anxiety and memory impairment [16]. Notably, acute and chronic social stress has been related to an alteration in NTs levels such as NGF and BDNF in humans and animals [41]. Recently, research proved a correlation between BDNF levels in the nucleus accumbens and the susceptibility to induced stress (e.g., defeat stimulation), causing social avoidance and anxiety-like and depression-like behaviours [42]. Moreover, even in a mouse model of post-traumatic stress disorder, in which the animals witnessed defeat events, the stress may provoke anxiety-like behaviours related to decreased BDNF levels in the hippocampus and medial prefrontal cortex and increased BDNF levels in the amygdala [43]. This evidence supports the role of the NTs in the stress-induced response mediated by the hypothalamus–pituitary–adrenocortical axis [41,43]. Finally, a reduction in neurotrophic support by BDNF during adulthood has been suggested by some authors in the development of neurodegenerative disorders such as Alzheimer’s, Parkinson’s, and Huntington’s diseases [8].

## 4. Neurotrophins and Lung

If the role of NTs in the nervous system is well known, only in the last years researchers showed an increasing interest in NTs’ implication in lung development. Accordingly, NTs appear to be involved in the physiology and pathophysiology of several airway diseases, neonatal lung diseases, allergic and inflammatory diseases, lung fibrosis, and even lung cancer (Table 2). Moreover, they have also been detected in other peripheral tissues, including immune cells, epithelium, smooth muscle, fibroblasts, and vascular endothelium [9].

### 4.1. Neurotrophins and Lung Development

Looking more specifically at lung development, NTs affect neuronal regulation of growth and proliferation of different lung elements by the following BDNF/TrkB, p75^NTR^, and BDNF.

BDNF/TrkB system is essential for the normal development of the lung and its extrinsic innervation. BDNF mRNA levels rise at 12 weeks of gestation, and TrkB mRNA at 13 weeks of gestation, in parallel with the start of lung development, and these levels are maintained until birth. Several experimental studies provided evidence that BDNF and NT-4 coordinate lung smooth muscle formation and innervation by extrinsic neuronal pathways [2,41]. The embryos of BDNF-knockout mice showed decreased axon branches and shortened axons targeting the smooth muscle without any change in lung morphogenesis [2,48]. Thus, BDNF serves as a target-derived neurotrophic factor for lung smooth muscle innervation by extrinsic neurons during embryogenesis [1]. Moreover, to ensure lung innervation, BDNF expression needs to be temporally coordinated with smooth muscle differentiation [49]. Indeed, BDNF mRNA is expressed as early as 11.5 days of gestation in the lung mesenchyme before its differentiation into smooth muscle fibre, and it is down-regulated after smooth muscle differentiation by a mechanism involving microRNA, miR-206 [2]. Moreover, BDNF−/− compared to NT4−/− homozygote knockout mice showed thinner bronchial epithelium, larger air space, lack of neuroepithelial bodies, and a significant reduction in nerve fibre density in the bronchial smooth muscle, submucous plexus in bronchioles, and pulmonary artery walls. In addition, airway innervation was reduced in the TrkB−/− mice than in the NT4−/− mice. BDNF knockout-animals experienced a reduction in the subset of afferents nerves involved in lung ventilator control and exhibited severe respiratory abnormalities characterised by depressed and irregular breathing and reduced chemosensory drive [3,44]. BDNF/TrkB is expressed in airway-related vagal preganglionic extrinsic neurons and plays a significant role in regulating cholinergic outflow to the airways. Cumulatively, these studies support the concept that NTs, secreted by the trachea and proximal pulmonary lobes, are responsible for extrinsic innervation [2,3].

During lung morphogenesis, p75^NTR^ consistently appears in non-neural cells adjacent to those expressing Trk receptors. The reciprocal patterns of expression indicate that the different localisation and activities of these two receptors most likely complement each other in regulating cell-cell interactions, which are essential for the innervation of lung non-neural tissue [2,60]. In embryonic rats, the bronchiole tubes are formed from 15.5 to 18.5 days of gestation. In this period, p75^NTR^ expression is high in the mesenchymal cells adjacent to the developing bronchiole epithelium. At this stage, the main bronchiole tubes are surrounded by neuronal cell adhesion protein (NCAM) and immunoreactive nerve fibre. However, these latter were not seen around the secondary branches, and NGF was not detected in the lung, suggesting that BDNF and/or NT3 could stimulate the Trk receptor on these cells. At embryonic day 16.5 of gestation, the nerve fibre ingrowth into the bronchiole area is observed. Interestingly, NCAM is also being expressed at low levels in the p75^NTR^-positive bronchiole mesenchymal populations. Therefore, expression of p75^NTR^ in the mesenchyme of the developing lung bronchioles preceded the ingrowth of nerve fibres and NCAM expression by 1.5–2 days. Since intrinsic neurons in both murine and human embryonic lungs express p75^NTR^, it is reasonable to assume that these nerve fibres represent both extrinsic and intrinsic neurons [1,2,3].

Other studies hypothesised that BDNF is a preferable ligand of α9-integrin. Moreover, this system can be involved in lung development. Integrins are heterodimeric transmembrane receptors regulating cell-cell and cell-extracellular matrix interactions [2,61]. Integrin α9β1 is expressed on a wide variety of cell types, interacts with many ligands, and has important functions in lung, lymphatic and venous valve development. α9 immunoreactivity in the murine lung was not detected until 12.5 days of gestation. The onset of α9 expression follows the rostral to caudal differentiation pattern of smooth muscle cells along the airways and coincides temporally and spatially with the expression of α-smooth muscle actin. Moderate staining levels of integrin α9 were seen in the cells surrounding the more developed proximal, primary bronchioles. The expression of α9 in airway smooth muscle rapidly increases to adult levels and remains high throughout the rest of the gestation [50]. This expression parallels the onset of tenascin (a natural extracellular matrix-ligand of α9) expression. It is initially restricted to the epithelial-mesenchymal interface and smooth muscle of the airways, and it is widely distributed throughout the mesenchyme of the nascent alveolar septa by 17.5 days of gestation. Since intense expression of α9 in distal adult human airway epithelium was also observed, but no tenascin immunoreactivity was measured, a study hypothesised that both in adult and embryo, BDNF is a preferable ligand of α9 integrin [2].

The temporal correlation in airway smooth muscle expression of BDNF, TrkB, p75^NTR^, and α9 supports their role in lung innervation by extrinsic autonomic nervous system neurons. Taken together, these findings are in line with the neurotrophic hypothesis and suggest that while several neurotrophins and Trk receptors are required for lung innervation, TrkB is more important than TrkA and TrkC in smooth muscle/epithelium differentiation and, together with p75^NTR^ and α9β1 are essential for lung development [1,2,3].

### 4.2. Neurotrophins Expression in the Peripheral Tissues

Several experimental evidence suggests that NTs and their receptors are expressed by a wide variety of lung cells, including immune cells, airway epithelium, airway smooth muscle, airway innervation, and pulmonary vasculature [1,2,3,4].

Lung monocytes and macrophages express mRNA and protein for NGF and BDNF, while interstitial macrophages express only BDNF and alveolar macrophages NT-3. Following the Immunoglobulin E cross-linking, NGF, BDNF and NT-3 are released by mast cells [62].

Overall, T lymphocytes constitutively express BDNF except for CD4+ and CD8+ T lymphocytes and Th2 cells, which release NGF. Conversely, following antigen stimulation, B cells produce NGF, BDNF and NT-3 [63].

Eosinophils have been shown to produce NGF, BDNF, and NT-3 at baseline and after NGF stimulation [64]. Following their release, NTs promote eosinophil survival and enhance IL-4 synthesis thanks to the expression of their receptors on the eosinophil surface [64].

Moreover, via non-genomic signalling, NTs may enhance or inhibit the secretion of pro- and anti-inflammatory cytokines, promote cell migration, and modulate the immune response. Similarly, via genomic signalling, they may influence the expression of cytokines and receptors in immune cells [65].

Sparse studies are available on the NTs expression in the airway epithelium [45,46]. Experimental evidence showed that NGF, BDNF, and NT-3 are constitutively expressed by the airway epithelium and, due to the TrkA expression in human bronchial epithelial cells, the latter may be a target of NTs [47]. The binding of NTs to their receptors induces Clara cell proliferation and repair of the lung tissue as well as nitric oxide (NO) production, thus, facilitating bronchodilation [47,66,67].

Airway smooth muscles are also a potential source of NTs. Immunocytochemical studies on human bronchial smooth muscle cells found that all NTs and their receptors TrkB and TrkC are constitutively expressed. These data support the evidence that airway smooth muscle produces NTs and is influenced by their action. In this regard, NTs modulate the airway smooth muscle function in a non-genomic manner, while via genomic mechanisms, they modulate cellular growth and proliferation [51].

Neural control of the airway involves three peripheral autonomic pathways: cholinergically mediated bronchoconstriction (involving the vagus), adrenergically mediated bronchodilation (adrenal medulla or sympathetic ganglia); and the non-adrenergic/non-cholinergic (NANC) pathway with both bronchodilatory and bronchoconstrictive components (involving parasympathetic efferent neurons, vasoactive intestinal peptide and NO, subepithelial C-fiber sensory afferents that release tachykinins) [68,69]. Data reported that NTs and receptors are expressed in lung innervation, where they promote de novo neuropeptide and tachykinin production, neuronal growth and remodelling, and neurogenic airway hyperreactivity [57,70,71].

Evidence shows that NTs signalling may control vascular structure and function [72,73,74]. In extrapulmonary branches of the human pulmonary artery, NGF, BDNF and NT-3 are expressed in the intima and adventitia, suggesting a local autocrine or paracrine effect of NTs on pulmonary vascular tone [73]. In the human pulmonary endothelial cells, BDNF and NT-3 can induce NO production and endothelial NO synthase (eNOS) expression resulting in acute vasodilation and endothelial cell proliferation and survival, respectively [74,75].

### 4.3. Neurotrophins and Lung Diseases

Given their pleiotropic effects, NTs have been recently investigated to assess their potential role in the pathophysiology of several airway disorders, such as neonatal lung, allergic, and inflammatory diseases, lung fibrosis and even lung cancer (Table 2).

Bronchopulmonary Dysplasia (BPD) is defined as a chronic lung disease affecting the infant born before 32 weeks of gestational age, with radiological evidence of damage to the lung parenchyma, and requiring respiratory support (either invasive or non-invasive) at 36 weeks of post-menstrual age at least for three or more consecutive days to maintain a peripheral arterial oxygen saturation (SpO2) > 90% [76,77]. BPD recognizes many prenatal risk factors, including maternal smoking, chorioamnionitis, intrauterine growth restriction, and postnatal risk factors such as patent ductus arteriosus, parenteral nutrition, sepsis, and mechanical ventilation [78].

Additionally, respiratory models showed the effects of hyperoxia on lung expression of BDNF, NGF, and Trk receptors in the development of BPD [79]. Compared with normoxic, age-matched old rat pups, seven days of continuous hyperoxia exposure increases BDNF levels and peribronchial smooth muscle TrkB expression, but it did not affect NGF and TrkA levels in the lung. These findings suggest that hyperoxic injury can contribute to the upregulation of NTs, thus, to airway reactivity featuring BPD [80]. In a similar animal model, authors reported that BDNF and TrkB signalling also participated in a centrally or peripherally induced increase in acetylcholine production and resulted in a cholinergic outflow to the airways, supporting the role of NTs in the central and peripheral neural control of the airways [80].

Asthma is a chronic inflammatory airway disease featured by bronchial smooth muscle thickening, aberrant matrix deposition, mucus overproduction, impaired airway microbiota, increased expression of inflammatory mediators, and reversible airway hyperreactivity [81,82,83]. In addition to the most common factors involved in asthma onset (e.g., exposures to allergens, respiratory viruses, cigarette smoke, ozone, and pollutants), NTs are also recognised as crucial players both in early life events and in long-term abnormalities occurring in the structure and function of the airway tree [57].

Several types of immune cells, such as mast cells, macrophages, dendritic cells, and lymphocytes, express NTs and their receptors, which, in turn, may modulate allergen sensitisation. Accordingly, allergen challenge has been related to an increase in NGF, BDNF and NT-3 levels, both in animal models of allergic asthma and in bronchoalveolar lavage fluid in humans [58,59].

Furthermore, NTs may influence airway hyperreactivity by inducing the release of inflammatory cytokines, which increase airway smooth muscle contractility [84].

Lastly, fibroblasts and bronchial smooth muscle are a target of the NTs action since the latter can modulate remodelling airway processes, including proliferation, migration, and secretion of inflammatory mediators [52].

Notably, there is evidence for the NTs’ role in the occurrence of pulmonary fibrosis. In vitro studies demonstrated that human pulmonary fibroblasts constitutively secrete NGF, which promotes mesenchymal cell proliferation and extracellular matrix components release, leading to pulmonary fibrosis [53,54]. This evidence was successively confirmed in a study involving adults suffering from idiopathic pulmonary fibrosis/usual interstitial pneumonia [55]. Fibroblasts derived from the enrolled patients showed higher proliferative rates in response to BDNF. Moreover, NGF-induced p75^NTR^ expression in human lung fibroblasts stimulated fibroblasts migration and increased collagen contraction [56]. However, these findings were not confirmed by another study in which NGF release was not associated with the survival of human lung fibroblasts nor stimulated TrkA expression [56].

Regarding the effects on tumour cells, it has been reported that NTs exert their action in the malignant transformation of blood and solid tumours both in autocrine/paracrine and indirect manner [85]. Specifically, cancer cells express NTs receptors and may produce NTs and respond to these mediators [86]. Accordingly, NGF has been found in bronchoalveolar and squamous cell carcinomas, while BDNF has been in small-cell lung cancers. Moreover, TrkA and TrkB have been identified in human lung adenocarcinoma, squamous cell carcinomas, and atypical carcinoids, and TrkB has been found in small-cell lung cancers and atypical carcinoids [86]. Interestingly, p75^NTR^ has not been detected in lung tumours, probably because its absence can contribute to cancer progression. Thanks to their angiogenic properties, NTs may indirectly modulate tumour growth: indeed, highly vascularised tumours express higher NTs levels, which promote cell growth and proliferation as well as maintain the tumour blood supply [87].

## 5. Discussion

NTs are a relatively recent discovery, and though most of the initial data in the literature focused on their role in the central and peripheral nervous system, evidence supports their involvement also in the development of different systems, particularly in the airway tract [1,2,8]. Their action is mediated by two types of receptors, with higher and lower affinity, which may interact with the mature or the immature form: the signalling cascades activated by these various combinations may result in opposite effects, giving to NTs a pleiotropic role in the evolving human organism [5]. In the nervous system, they exert their function both in the developing and mature brain, participating in cell migration and proliferation, synapse formation, dendritic outgrowth regulation, and axonal orientation [18,19,20]. In the pulmonary system, NTs participate in the neuronal regulation of growth and proliferation of different lung elements, coordinating the lung smooth muscle formation and the innervation by extrinsic neuronal pathways [2,48]. Moreover, NTs are produced by immune cells and are involved in the expression of cytokines and receptors on the immune cells’ surface [63,64].

Several studies on animal models proved that NTs and their receptors are expressed since the early stages of the embryo’s development and, recently, their role in the precocious phases of growth was confirmed by analyses performed on women’s blood during all the course of the pregnancy [4,24,27,29,30,31]. Therefore, a disruption of their function may induce clinical manifestations in one or more systems: an alteration in serum levels of NTs has been related to a pregnancy complication, intrauterine growth impairment, and poor neurodevelopmental outcomes [20,27,29,30,31]. Later in life, altered levels of NTs have been demonstrated in chronic pathologies of the nervous system (e.g., epilepsy, schizophrenia, depression, and neurodegenerative diseases) and the pulmonary system (e.g., asthma, allergic and inflammatory diseases, lung fibrosis, and lung cancer) [1,8].

Particularly, BDNF has been extensively studied as the neurotrophic factor with a major role in nervous and pulmonary diseases [1,2,8]. The alteration in its expression has been linked to neurodevelopmental disorders with an early onset and severe clinical manifestations, often named “synaptopathies” because of structural and functional synaptic plasticity abnormalities [38,88,89]. A reduction in BDNF levels has been demonstrated in neonatal blood samples of autistic subjects and in numerous animal models of autism and genetic syndromes related to autistic behaviours (e.g., Rett syndrome), but also in patients affected by schizophrenia and depression [6,16,33,34,38,90]. On the contrary, increased BDNF levels have been detected in the blood of patients affected by intellectual disability (ID) [88]. Thus, interesting data is represented by the higher BDNF levels in subjects affected by ASD and ID when compared to cases without ID, explaining the controversial higher levels of BDNF in autistic children reported in some studies [39,40]. Likewise, results on ADHD subjects are inconsistent. Some studies reported increased serum BDNF levels in ADHD patients compared to healthy subjects, while other authors refuted these results, proving reduced or similar BDNF levels in patients and controls [32,38,89]. The discrepancy among the data regarding these pathologies may be ascribed to methodological differences in assay methods, time of collection and preparation of the samples, and patient characteristics such as the number of cases and controls, gender, inclusion and exclusion criteria, and subgroup division. In addition, some confounding factors, such as intellectual disability, being overweight, and intense physical activity, have not been accounted for in several papers.

Few researchers have also considered allergies a confounding feature due to the recent report of elevated BDNF levels in patients with allergic diseases, such as asthma [38]. Different cellular components of the lung, including epithelium, smooth muscle and immune cells, show NTs and receptor expression. The latter may modulate cell proliferation, promote apoptosis, affect the immune response within the lung, and influence neural control of the airway, modulating the balance between bronchoconstriction and bronchodilation. Taking advantage of the evidence for NT signalling in airways, there is currently significant interest in the potential role of NTs in asthma [1]. Allergen challenge is associated with a significant increase in humans’ NGF, BDNF and NT3 levels in bronchoalveolar lavage fluid [59]. Several immune cells, such as mast cells, macrophages, dendritic cells, and lymphocytes, express NTs and their receptors [1]. NTs may influence neural control of the airway. By modulating the NO release, NTs affect epithelium-derived bronchodilator response and airway hyperreactivity [67]. Lastly, NTs may influence airway remodelling by increasing fibroblast differentiation, migration and differentiation [58].

## 6. Conclusions

Recently, many studies have investigated the role of NTs in the onset of several human pathologies, identifying both expected and unexpected functions for this protein family and their receptors. The NTs have a crucial role in the survival and differentiation of both neuronal and airway cell populations during development. Whether in the brain, NTs may maintain high expression levels and influence both inhibitory and excitatory synaptic transmission and activity-dependent plasticity; in the lung, NTs are crucial for normal lung development as well as developmental lung disease, by working at different levels thanks to the presence of a large amount of data several cell systems.

Herein, the above-reported discussion on the NTs’ role and functions leads to the conclusion that they are involved in the physiology and, consequently, in the dysfunction of the nervous system and airway tree, also appearing as a crucial convergence point for the brain–lung axis (Figure 2). In light of their pleiotropic role and considering the data reviewed here, further studies are urgently required to confirm the NTs as reliable biomarkers that relate specifically to the development, occurrence, and progression of the neurologic and pulmonary pathological conditions.

## Figures and Tables

**Figure 1 ijms-24-07089-f001:**
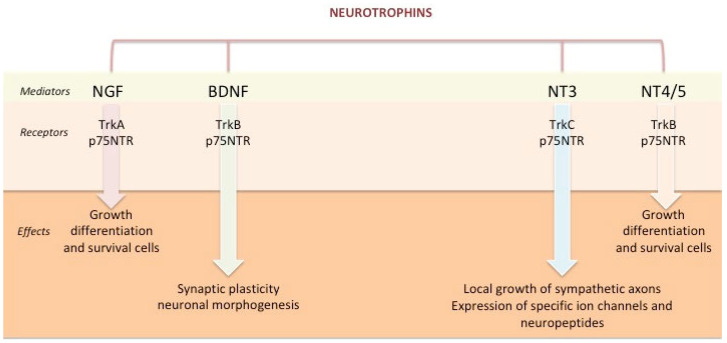
Neurotrophins: Role and effects. Legend: NGF = nerve growth factor; BDNF = brain-derived neurotrophic factor; NT = neurotrophin; Trk = tropomyosin-related kinase P75^NTR^ = P75 neurotrophin receptor.

**Figure 2 ijms-24-07089-f002:**
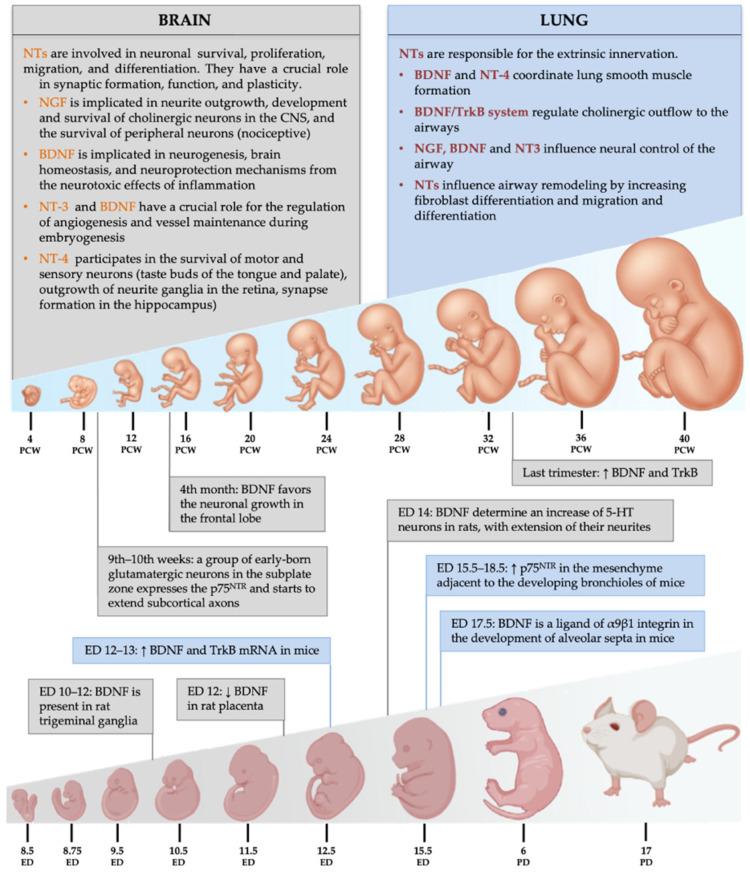
Schematic representation of NTs’ role during pregnancy in humans and rodents. The upper part of the figure shows the role of NTs in the development of the brain and the lungs. In the lower part of the figure, the boxes display the involvement of the NTs at different gestational ages in human and rodent embryos/fetuses. Legend: BDNF = brain-derived neurotrophic factor; ED = embryonic day; NGF = nerve growth factor; NT = neurotrophin; PCW = post-conceptional week; PD = postnatal day; p75^NTR^ = p75 neurotrophin receptor; Trk = tropomyosin-related kinase; 5-HT = serotonin; ↓: decrease; ↑: increase.

**Table 1 ijms-24-07089-t001:** NTs/receptors interactions and implications in brain diseases.

Neurotrophin	Receptor	Brain Region	Effects	Implication in Diseases	References
BDNF	TrkB	Hippocampus	Promotion of mossy fibers sprouting of the dentate granule cells	↓ BDNF in deficits of learning and memory ↓ BDNF in ASD ↑ BDNF in seizure activity in temporal lobe epilepsy	[8,16,30,32,36,38,39,40]
	TrkB/ P75^NTR^	Brainstem and cerebellum	Maturation of GABAergic neurons and development of synapses	↓ BDNF in RTT syndrome	
	NA	HPA axis	Survival and homeostasis of dopaminergic neurons	Controversial data on BDNF in ADHD	
	TrkB/ P75^NTR^	Frontal cortices and hippocampus	Survival, proliferation, differentiation, and synaptic plasticity	↓ BDNF in schizophrenia ↓ BDNF in depression	
	TrkB	CNS	Survival, proliferation, differentiation, and synaptic plasticity	↑ BDNF in neonatal blood of children with ASD and ID	
NGF	TrkA	Basal forebrain and prefrontal cortex	Survival of cholinergic neurons and involvement in learning processes and attention systems	↑ NGF in ADHD	[37]
	TrkA/ P75^NTR^	CNS	Homeostasis of mitochondria	↑ NGF in ASD	
NT-3	NA	Cerebellum	Survival of neurons and cerebellar growth	↑ NT-4 in ASD	[27]
NT-4	TrkB	CNS	Survival, proliferation, differentiation, and synaptic plasticity	↑ NT-4 in neonatal blood of children with ASD and ID	[30]

Legend: ADHD = Attention Deficit/Hyperactivity Disorder; ASD = Autism Spectrum Disorders; BDNF = brain-derived neurotrophic factor; CNS = central nervous system; HPA = Hipotalamo-pituitary-adrenal; ID = intellectual disability; NA = not available; NGF = nerve growth factor; NT = neurotrophin; P75^NTR^ = P75 neurotrophin receptor; RTT = Rett; Trk = tropomyosin-related kinase; ↓: decrease; ↑: increase.

**Table 2 ijms-24-07089-t002:** Neurotrophin targets, effects, and implications in airway diseases.

Airways Elements/Tissue	Targets	Effects	Implication in Diseases	References
Bronchial epithelium	TrkB P75^NTR^	Lung development Inhibit apoptosis and/or promote cell survival Airway remodeling NO release	↑ BDNF in asthma ↑ BDNF and TrkB, ↓ p75^NTR^ with infection ↑ Airway thickening in inflammation/asthma ↑ TrkB, p75^NTR^ with cigarette smoke	[44,45,46,47]
Airway smooth muscle	TrkB P75^NTR^	Airway development and remodeling Increased airway contractility Cell proliferation	↑ BDNF, TrkB with hyperoxia in fetal/neonatal ASM ↑ P75^NTR^ and TrkB in inflammation/allergic diseases ↑ BDNF in airway inflammation ↑ ECM deposition ↑ TrkB, p75^NTR^ with cigarette smoke ↑ ASM contractility with cigarette smoke and in inflammation	[1,2,3,41,48,49,50,51]
Nerves	TrkB P75^NTR^	NK1 receptor expression Neurotransmitter release Ach content of sensory nerves	↑ BDNF in asthma ↑ Neuroplasticity ↑ Release of cholinergic neurotransmitters ↑ Neuromediated airway hyperresponsiveness	[3,44]
Fibroblasts	TrkB P75^NTR^	Proliferation Migration Differentiation	↑ BDNF and TrkB in inflammation	[52,53,54,55,56]
Blood	ND	ND	↑ BDNF in asthma ↑ p75^NTR^ with cigarette smoke	[1,8,38,57]
BAL	ND	ND	↑ BDNF in asthma	[58,59]

Legend: BDNF = brain-derived neurotrophic factor; NGF = nerve growth factor; NT = neurotrophin; P75NTR = P75 neurotrophin receptor; ECM = extracellular matrix; ASM = Airway smooth muscle; Trk = tropomyosin-related kinase; NK1 = Neurokinin 1; Ach = acetylcholine; NO = nitric oxide; ND = not detected; ↓: decrease; ↑: increase.

## Data Availability

Not applicable.

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
