# Peer review of "Neurotrophins: Expression of Brain–Lung Axis Development"

_ijms, 2023, doi:10.3390/ijms24087089_

Round 1

Reviewer 1 Report

Review of Manuscript “Neurotrophins: expression of brain-lung axis development” submitted to the International Journal of Molecular Sciences by Sara Manti et al.

This is an interesting review focused on the role of neurotrophins in the central and peripheral nervous system as well as in lung development. This is a new topic with increased interest and enhanced data in the last decade. The manuscript has presented clinical and experimental data about the physiological and pathophysiological role of neurotrophins in brain and lung development. The manuscript reveals the mechanisms of action of different neurotrophins, their receptors, and the following effects on both systems. Furthermore, the authors have presented the role of neurotrophins in various brain and lung diseases, where the levels of these molecules can be increased or decreased. Moreover, the levels of neurotrophins vary during the different trimesters in both systems. Tables and figures provide good and detailed information which helps to understand the complicated processes. The paper is very well-written, easy to read and follow the presented information. In this line, it could be published the way it is.

My minor recommendations are listed below:

1.      Line 347-348 the abbreviation ASM = Airway smooth muscle is given twice.

2.      Line 455 at the end of the sentence some references should be included supporting the presented information.

Author Response

Dear Editor/Collegues,

thank you for your letter concerning our manuscript entitled “Neurotrophins: expression of brain-lung axis development” for publication in International Journal of Molecular Sciences.

We are extremely grateful to the Referees for the helpful remarks on our manuscript.

We have modified the text according to the request. Changes have been highlighted at the particular points where it was revised.

Some sentences have been rephrased. Three figures have been added.

The final manuscript has been seen and approved by all authors.

Reviewer 1:

This is an interesting review focused on the role of neurotrophins in the central and peripheral nervous system as well as in lung development. This is a new topic with increased interest and enhanced data in the last decade. The manuscript has presented clinical and experimental data about the physiological and pathophysiological role of neurotrophins in brain and lung development. The manuscript reveals the mechanisms of action of different neurotrophins, their receptors, and the following effects on both systems. Furthermore, the authors have presented the role of neurotrophins in various brain and lung diseases, where the levels of these molecules can be increased or decreased. Moreover, the levels of neurotrophins vary during the different trimesters in both systems. Tables and figures provide good and detailed information which helps to understand the complicated processes. The paper is very well-written, easy to read and follow the presented information. In this line, it could be published the way it is.

My minor recommendations are listed below:

  1. Line 347-348 the abbreviation ASM = Airway smooth muscle is given twice.

R: as suggested, we deleted the duplicated abbreviation

  1. Line 455 at the end of the sentence some references should be included supporting the presented information.

R: as suggested, we added reference to support the reported information

Reviewer 2 Report

In the review article ‘Neurotrophins: expression of brain-lung axis development’, the authors nicely compiled the latest information about Neurotrophins and their role in lung development. I would suggest adding a descriptive figure which can reflect the pathways and downstream targets of Neurotrophins.

Author Response

Dear Editor/Collegues,

thank you for your letter concerning our manuscript entitled “Neurotrophins: expression of brain-lung axis development” for publication in International Journal of Molecular Sciences.

We are extremely grateful to the Referees for the helpful remarks on our manuscript.

We have modified the text according to the request. Changes have been highlighted at the particular points where it was revised.

Some sentences have been rephrased. Three figures have been added.

The final manuscript has been seen and approved by all authors.

Reviewer 2:

In the review article ‘Neurotrophins: expression of brain-lung axis development’, the authors nicely compiled the latest information about Neurotrophins and their role in lung development. I would suggest adding a descriptive figure which can reflect the pathways and downstream targets of Neurotrophins.

R: as requested, a descriptive figure has been added to the manuscript. Please, see figure 1.